# Genome Assembly and Analysis of the Flavonoid and Phenylpropanoid Biosynthetic Pathways in Fingerroot Ginger (*Boesenbergia rotunda*)

**DOI:** 10.3390/ijms23137269

**Published:** 2022-06-30

**Authors:** Sima Taheri, Chee How Teo, John S. Heslop-Harrison, Trude Schwarzacher, Yew Seong Tan, Wei Yee Wee, Norzulaani Khalid, Manosh Kumar Biswas, Naresh V. R. Mutha, Yusmin Mohd-Yusuf, Han Ming Gan, Jennifer Ann Harikrishna

**Affiliations:** 1Centre for Research in Biotechnology for Agriculture, University of Malaya, Kuala Lumpur 50603, Malaysia; sima.taheri100@gmail.com (S.T.); cheehow.teo@um.edu.my (C.H.T.); yusmin_y@um.edu.my (Y.M.-Y.); 2Department of Genetics and Genome Biology, University of Leicester, Leicester LE1 7RH, UK; ts32@leicester.ac.uk (T.S.); mkb35@leicester.ac.uk (M.K.B.); 3Key Laboratory of Plant Resources Conservation and Sustainable Utilization/Guangdong Provincial Key Laboratory of Applied Botany, South China Botanical Garden, Chinese Academy of Sciences, Guangzhou 510650, China; 4Institute of Biological Sciences, Faculty of Science, University of Malaya, Kuala Lumpur 50603, Malaysia; tyewseong@yahoo.com; 5School of Science, Monash University Malaysia, Subang Jaya 47500, Malaysia; wee.weiyee@monash.edu; 6Department of Biology, International University of Malaya-Wales, Kuala Lumpur 50603, Malaysia; norzulaani@iumw.edu.my; 7Division of Infectious Diseases, Vanderbilt University Medical Center, Nashville, TN 37203, USA; nareshmvr@gmail.com; 8Biology Division, Centre for Foundation Studies in Science, University of Malaya, Kuala Lumpur 50603, Malaysia; 9Department of Biological Sciences, Sunway University, Bandar Sunway, Petaling Jaya 47500, Malaysia; hxg2760@gmail.com

**Keywords:** *Boesenbergia rotunda*, DNA methylation, genome assembly, ginger, panduratin A, SSR, TE

## Abstract

*Boesenbergia rotunda* (Zingiberaceae), is a high-value culinary and ethno-medicinal plant of Southeast Asia. The rhizomes of this herb have a high flavanone and chalcone content. Here we report the genome analysis of *B. rotunda* together with a complete genome sequence as a hybrid assembly. *B. rotunda* has an estimated genome size of 2.4 Gb which is assembled as 27,491 contigs with an N50 size of 12.386 Mb. The highly heterozygous genome encodes 71,072 protein-coding genes and has a 72% repeat content, with class I TEs occupying ~67% of the assembled genome. Fluorescence in situ hybridization of the 18 chromosome pairs at the metaphase showed six sites of 45S rDNA and two sites of 5S rDNA. An SSR analysis identified 238,441 gSSRs and 4604 EST-SSRs with 49 SSR markers common among related species. Genome-wide methylation percentages ranged from 73% CpG, 36% CHG and 34% CHH in the leaf to 53% CpG, 18% CHG and 25% CHH in the embryogenic callus. Panduratin A biosynthetic unigenes were most highly expressed in the watery callus. *B rotunda* has a relatively large genome with a high heterozygosity and TE content. This assembly and data (PRJNA71294) comprise a source for further research on the functional genomics of *B. rotunda*, the evolution of the ginger plant family and the potential genetic selection or improvement of gingers.

## 1. Introduction

*Boesenbergia rotunda* (L.) Mansf. (*syn. B. pandurata* (Roxb.) Schltr.) (ITIS Taxonomic Serial No.: 506504), commonly known as Fingerroot ginger and as a type of galanga or galangal, is a member of the family Zingiberaceae in the order Zingiberales. The common name for the plant, Fingerroot, is due to its finger-shaped rhizomes (Figure 1B). With 50 genera and 1600 species, the Zingiberaceae is the largest family in the order, along with other families of ginger (Zingiberaceae, Costaceae, Marantaceae, and Cannaceae) and banana (Musaceae, Strelitziaceae, Lowiaceae, and Heliconiaceae), that include many economically important plant species [1,2]. The Zingiberaceae family consists of herbaceous perennial plants distributed over tropical and subtropical regions with the highest diversity in Southeast Asia (especially in Indonesia, Malaysia and Thailand), India and Southern China [3,4,5,6]. The leaves, flowers and in particular the rhizomes of many of the Zingiberaceae family members are used as flavouring agents and for herbal medicine [4,7]. 

*Boesenbergia* is a genus that has around 80 different species and can be found all the way from India to Southeast Asia [3,8,9,10]. *B. rotunda* is a perennial herb propagated via rhizomes and widely cultivated commercially for its rhizomes and shoots to flavour food and for ethno-medicinal use [11,12]. Antioxidant activity is among the most important bioactivities of plant flavonoid compounds [13], and research on the secondary metabolites of *B. rotunda* has focused on the medicinal properties of the rhizome extracts, in particular the flavanones and chalcones including panduratin A, pinocembrin, pinostrobin, alpinetin, boesenbergin, cardamonin, naringenin, quercetin, and kaempferol [8,14,15,16,17,18,19,20]. Of these, the flavonoid compounds, panduratin A/DI, 4-hydroxypanduratin, and cardamonin show the clearest biological and pharmacological effects such as being anti-inflammatory [21,22], anti-tumour activity against human breast and lung cancers [23,24,25,26,27], and antimicrobial activity against HIV protease [28], Dengue-2 (DEN-2) virus NS3 protease [29,30], SARS-CoV-2 in human airway epithelial cells [31], the oral bacteria *Streptococcus mutans* [32,33], *Helicobacter pylori* [34], and against the spoilage bacteria *Lactobacillus plantarum* (Lactiplantibacillus) [35]. A recent patent claimed that panduratin derivatives from *B. rotunda* have the potential for preventing, ameliorating, or treating bone loss disease [36], while 4-hydroxypanduratin was reported to have the most potent vasorelaxant activity among the major flavonoids of the *B. rotunda* extracts [37]. 

The ethno-medicinal and potential pharmaceutical importance of *B. rotunda* have led to interest in exploring cell and tissue cultures for secondary metabolite production. In commercial farms, the plant is propagated clonally from rhizomes, and several protocols for multiplication via in vitro culture have been reported including plantlet regeneration via somatic embryogenesis from callus cultures [38,39], shoot bud explants [40] and embryogenic cell suspension cultures [38]. Cell suspensions of *B. rotunda* [18,41] and various types of callus [42,43] have been explored as potential sources for alpinetin, cardamonin, pinocembrin, pinostrobin and panduratin A. Reproducible methods for the in vitro cell culture of *B. rotunda*, have led to protocols for genetic transformation [39], that could facilitate the metabolic engineering of cell materials for specific desirable metabolite production; however, current knowledge of the underlying biosynthetic pathways is sparse. Other than biochemical profiling [17,43], the application of current technologies for determining the deep sets of the genetic sequences expressed in various tissue and cell types can deliver useful information. 

Genomic level studies improve the understanding of the biology and biochemistry of the plant and can be applied in breeding for improved agronomy and plant products. Whole genome sequencing identifies the genes and regulatory sequences for complex biological processes such as secondary metabolite biosynthesis [44,45,46], while transcriptional profiling provides information for functional studies. Structural genomic studies have been undertaken in other Zingiberales including turmeric (*Curcuma longa*; genome size of 1.24 Gb) [47] and for several Musaceae species and cultivars, which have genome sizes ranging from 462 Mb to 598 Mb (Banana Genome Hub https://banana-genome-hub.southgreen.fr/ accessed on 28 May 2022) [48], while the Pan-genome of *Musa Ensete* has a genome size of 951.6 Mb [49]. Larger plant genomes have now been sequenced including those of important monocot species such as wheat ~17 Gb (International Wheat Genome Sequencing Consortium), *Aegilops tauschii* ~4.3 Gb [50], oil palm ~1.8 Gb [51] and maize ~2.6 Gb [52], in addition to species known for their unique metabolites such as tea (*Camellia sinensis*) ~ 2.98 Gb [53,54] and ginseng (*Panax ginseng*) ~3.2 Gb [55]; however, even with the recent advances in long sequence technology, large plant genomes can be challenging to assemble due to a high repeat content and high levels of heterozygosity [56,57].

The availability of an assembled genome sequence expands the functional biological questions that can be asked, since regulatory and variable elements, many of which may be involved in epigenetic regulation, cannot be seen purely using expression data. Consequently, while transcriptome [41] and proteome [58] data for *B. rotunda* are available, the lack of a previously published genome assembly is a limitation for functional studies. Genome assemblies also facilitate the exploration of genomic repeats which can not only be a source for genetic markers but are also drivers of genome size, gene content and order, centromere function and reflect genome evolution [59,60]. Last but not least, the epigenetic dynamism in genomes mainly involves “non-coding DNA”, thus, a genome assembly provides the framework for epigenetic studies; therefore, in the current investigation we performed the first complete genome sequence for *B. rotunda* made with a hybrid assembly strategy using the Pacific Biosciences (PacBio) and Illumina HiSeq platforms. We explored the sites of 45S rDNA and 5S rDNA on metaphase chromosomes observed by Fluorescence in situ hybridization (FISH). In addition, we carried out a deep transcriptome (RNA-seq data) assembly from five *B. rotunda* samples, including various types of callus cultures, and leaves. Gene expression profiles and bisulfite seq DNA methylation data from these tissues and samples were used for a co-expression analysis to identify any association of gene expression and local DNA methylation of unigenes related to methylation, somatic embryogenesis, and the pathways for flavonoid and phenylpropanoid biosynthesis. We also report novel expressed sequence tags-SSR (EST-SSR) and genomic SSR markers for *B. rotunda* and the estimated cross-transferability of the designed primers between *B. rotunda* and closely related species to provide deeper genetic resources to support further study of the biology and biodiversity in this genus. Genomic information and complete sequence data for this less investigated herb should provide a solid foundation as a vital step in genetic analysis to facilitate *B. rotunda* improvement and to reach a deeper understanding of the metabolic pathways of its natural products. 

## 2. Results

### 2.1. Chromosomes and Location of rDNA Sites

*Boesenbergia rotunda* (2n = 36; 18 pairs of submetacentric chromosomes) has three pairs of 45S rDNA sites near the ends of three pairs of chromosomes (Figure 1A). One pair of the 5S rDNA sites (Figure 1D) is on a chromosome pair not bearing 45S rDNA.

### 2.2. Genome Assembly

Genomic DNA from the leaves of a single, clonal *B. rotunda* plant was sequenced using multiple approaches (Appendix A), with 114 Gb PacBio long reads, 260 Gb of Illumina HiSeq 2500 250 bp paired-end reads, and 90 Gb of mate-paired reads with 2, 5, 10, 20 and 40 kb insert sizes. Based on the k-mer analysis (k = 17, GenomeScope), the estimated haploid genome size of *B. rotunda* was 2.4 Gb (Appendix A), consistent with the flow cytometry (Appendix A). The heterozygosity was estimated as 3.01%. A hybrid genome assembly pipeline combining the Illumina data and PacBio data was adopted (Appendix A). The final assembled genome size was 2.347 Gb, characterized by 27,491 contigs and 10,627 scaffolds, with a contig N50 of 123.86 kb and a scaffold N50 of 394.68 kb (Table 1). Based on a benchmarking universal single-copy orthologs (BUSCO) analysis [61] mapping the *B. rotunda* genome against a set of 1440 core eukaryotic genes, 1232 (85.6%) were present (Appendix A). An assembly quality assessment showed that over 95% of the Illumina PE250 reads were mapped to the contig assembly (Table 2).

### 2.3. Annotation of the B. rotunda Genome

Five sets of RNA-seq datasets were generated from three cell culture types, in vitro and ex vitro leaves of *B. rotunda*, given the importance for secondary metabolites production. Individual transcriptomes were assembled from these RNA-seq reads using different de novo transcriptome assemblers (Table 3, Appendix A). The assembled transcriptome size ranged from 31 to 71 million base pairs with 72,085 to 158,465 contigs for the Oases, SOAPdenovo-Trans, TransAbyss, and Trinity (Table 3, Appendix A). Oases had the largest N50 size and average contig length. The BUSCO quantitative measure of the completeness transcriptomes in terms of expected gene content scores, also showed Oases (36.7%) and TransAbyss (36.6%) to give the assemblies with higher numbers of complete and single-copy contigs compared to the SOAPdenovo-Trans and Trinity (31.7%) (Appendix A). The non-redundant transcript sequences formed from the Oases followed by TGICL were used to annotate the *B. rotunda* genome and for the downstream expression analysis.

Based on a combination of de novo and homology-based gene prediction methods, 72.51% of the genome (1.70 Gb) was annotated as repeats including 6.94% tandem repeats. Among the Class I TEs (Retroelements), long terminal repeats (LTRs) constituted the greatest proportion of the genome (67.16%) while DNA TE made up 3.29 % of the genome (Appendix A, Table 4). 

From the 10,627 assembled contigs and 95,847 assembled transcriptome sequences searched for SSRs, (Table 5, Figure 2), the density of the microsatellites was 102 SSR loci per Mbp in the genomic and 69 SSR loci per Mbp in the transcriptome sequences. Among the identified repeat motif types, trinucleotides were the most abundant in both genomic (35.62%) and transcriptome (51.67%) sequences, followed by mono- and dinucleotide repeats (Table 5, Figure 2a). The class II type SSR-loci (<30 bp) were two-fold higher than the class I type in the genomic sequences, whereas the class II type SSR-loci were four-fold higher than the class I type SSR loci in the transcriptome sequences (Figure 2b). The number of AT rich microsatellites was significantly higher than that of the GC rich and microsatellites with a balanced GC content.

The mapping of *B. rotunda* SSR to close relatives using newly designed primer sequences showed that from 93.81% of the genomic SSR and 73.12% of the transcriptome sequences suitable for the SSR primer design, only a low number of primers mapped to the selected relatives, *Musa acuminata*, *Musa balbisiana*, *Musa itinerans* and *Ensete ventricosum* (Table 5). Overall, 224 G-SSR and 65 EST-SSR primers showed transferability into any of the four related species (with slightly more in the Ensete), while only 42 genomic SSRs and 7 transcript SSRs were common to all five genomes (Figure 2c,d). A subset of 14 *B. rotunda* SSR primer pairs (Appendix A) were tested for their marker potentiality and all showed amplified bands of the expected sizes for each species (Appendix A).

The annotation of the predicted protein-coding genes was a combination of homology-based and de novo prediction in addition to comparison with the *B. rotunda* transcriptome data (Appendix A). After consolidation, 73,102 protein-coding genes were predicted in the *B. rotunda* genome with an average transcript length of 4312 bp (excluding UTR), CDS length of 1360 bp, average exon and intron lengths of 303 bp and 812 bp, respectively, and 4.49 exons per gene (Appendix A). For the homology-based protein-coding gene predictions, the protein sequences from four species (*M. acuminata*, *Phoenix dactylifera*, *Oryza sativa* and *Arabidopsis thaliana*) were mapped onto the *B. rotunda* genome. From these alignments, *B. rotunda* had the highest number of matches with *P. dactylifera* followed by *O. sativa*, *A. thaliana* and *M. acuminata* (Appendix A). Functional annotation of the 73,102 predicted proteins from *B. rotunda* against seven databases enabled functional predictions for 97.8% of the predicted genes (Table 6). Non-coding RNA analysis of the assembly identified 213 microRNA (miRNA), 2727 transfer RNA (tRNA), 486 ribosomal RNA (rRNA), and 2136 small nuclear RNA (snRNA) genes (Table 7).

A final genome annotation was performed by using MAKER together with de novo assembled non-redundant transcripts, predicted proteins, non-coding RNAs and repeats.

### 2.4. Functional Classification by Gene Ontology

From a total of 95,847 unigenes derived from the *B. rotunda* transcriptome, 41,550 unigenes (43.35%) were found significantly scoring BLASTX hits against the NR protein database. Of these, 6850 (7.15% of the total unigenes) returned significant sequence alignments that could not be linked to Gene Ontology entries; 6038 (6.3%) of the GO mapped dataset did not obtain an annotation assignment and we could assign functional labels to 28,662 (29.9%) of the input sequences (Appendix A). Species distribution among the BLASTX matches showed that *M. acuminata* subsp. malaccensis had a very high similarity score with 87,000 top BLASTX hits from *B. rotunda*. Other species matches included *Ethiopian banana*, *Ensete ventricosum* (Musaceae) with 70,000 hits, African oil palm, *Elaeis guineensis* (Arecaceae) with 62,500 BLASTX hits and date palm, *P. dactylifera* (Arecaceae) with 62,000 BLASTX hits (Appendix A). Biological processes were the most represented functional group based on the categorization of the GO classes for the Nr annotated sequences (Figure 3a). 

A Blast2GO enzyme code (EC) annotation showed the distribution of *B. rotunda* predicted proteins among six main enzyme classes of oxidoreductases (1400), transferases (3500), hydrolases (2250), lyases (450), isomerases (250), and ligases (270) (Appendix A). The KOG function classification produced Nr hits for 18,767 unigenes annotated and classified functionally into 25 KOG functional categories (Figure 3b). General function prediction was the most represented group (2396) followed by signal transduction mechanism (2178) and posttranslational modification, protein turnover, and chaperons’ with 2031 genes. Comparison of all the *B. rotunda* unigenes with the KEGG pathway database resulted in 1494 out of 28,662 annotated unigene sequences (5.21%) being assigned to 145 predicted metabolic pathways.

### 2.5. Phylogenetic Orthology Inference of B. rotunda Genes

A total of 62,520 orthogroups were found with Orthofinder [62] (Appendix A) with matches of the genes from *B. rotunda* to 979,315 genes from 13 other species (*Glycine max*, *Cucumis melo*, *Gossypium raimondii*, *Brassica napus*, *Arabidopsis thaliana*, *Solanum tuberosum, Solanum lycopersicum*, *Musa acuminata*, *Zea mays, Oryza sativa* subsp. japonica, *Hordeum vulgare*, *Phoenix dactylifera* and *Brachypodium distachyon*). Of these, 7276 orthogroups were shared among all species and there were no single copy orthogroups (Appendix A). The species tree inferred by STAG [63] and rooted by STRIDE [64] indicated that *B. rotunda* has the closest relationship with M. acuminata (order Zingiberales) and *P. dactylifera* (order Arecales) followed by members of the Poaceae family (*Z. mays*, *O. sativa* subsp. japonica, *H. vulgare*, and *B. distachyon*), and was distant from the plant species from the Solanaceae, Brassicaceae, Malvaceae, Cucurbitaceae, and Fabaceae (Figure 4a, Appendix A). UpSet plotting showed 7276 orthogroups shared between the *B. rotunda* and 13 selected reference genomes (Figure 4b). Additionally, 1849 protein orthologs are specific for *B. rotunda* and 274 orthogroups are shared among the 13 selected reference genomes except for *B. rotunda*.

#### Gene Family Expansion and Contraction

Using the data generated from OrthoFinder [62], we explored the gene family expansion and contractions in *B. rotunda* (Figure 4a). In total, there are 17,106 gene families shared by the most recent common ancestor (MRCA). There were large numbers of gene families expanding (53–10,855) or contracting (16–11,754) between 14 plant genomes (Figure 4a). Our results show the substantial expansion of gene families in the Poaceae (5557) followed by the Brassicaceae (5104) and the Pooideae subfamilies (4205). A large gene family contraction was observed in Solanaceae (8975). Interestingly, the majority of the genomes with reported ancient whole genome duplication or massive segmental duplications or major chromosomal duplications show a higher number of gene family duplications than gene family losses (indicated by asterisks in Figure 4a).

### 2.6. Transcriptome Changes of B. rotunda Unigenes Related to Flavonoid and Phenylpropanoid Biosynthesis Pathways

A Transcriptome analysis showed that, in total, 167 unigenes from *B. rotunda* were mapped to five different classes of enzymes including oxidoreductase, transferase, ligase, lyase, and hydrolase in the flavonoid and phenylpropanoid pathways. Of these, only 23 enzymes showed differential expression in the different samples, i.e., in vitro leaf (IVL), embryogenic callus (EC), and non-embryogenic calli (dry callus (DC) and watery callus (WC)) using ex vitro leaf (EVL) samples as the comparator (Figure 5, Appendix A). Phenylalanine ammonia-lyase (PAL), the first enzyme in the phenylpropanoid pathway, turns phenylalanine to cinnamic acid. The PAL was expressed at the lowest levels among all the samples in the IVL, with the highest expression level in the WC (indicated by the dark red squares in Figure 5). Then, the coenzyme A (CoA) would attach to cinnamic acid or p-coumaric acid by 4-coumarate–CoA ligase (4CL) and form cinnamoyl-CoA or p-coumaroyl-CoA. This enzyme showed a relatively higher expression in all the samples except for the EC. 

In the phenylpropanoid pathway, cinnamic acid is also converted to coumarinate by Beta-glucosidase (BGLU) to produce coumarin. BGLU was expressed in all samples, with the highest expression level in the non-embryogenic calli (NEC). Then CHS, chalcone synthase (CHS) converts cinnamoyl-CoA to pinocembrin chalcone and p-coumaroyl CoA to naringenin chalcone. CHS was expressed in all the samples except the IVL, with the highest expression level in the WC. In the next step, the two flavanones of pinocembrin and naringenin are synthesised by chalcone isomerase (CHI). CHI was expressed in all the samples except the IVL, with the highest expression level in the EC. Pinocembin is converted to pinostrobin by flavanone-3-hydroxylase (F3H) which serves as a precursor of panduratin A synthesis. Expression analysis of the unigenes related to the F3H enzyme and dihydroflavonol 4-reductase (DFR), which are involved in the synthesis of anthocyanidins such as pelargonidin, cyanidin, and delphinidin, showed the DFR to be more highly expressed in the DC and WC compared to the other samples, while the F3H was only relatively up-regulated in the WC. Other enzymes in the phenylpropanoid pathway include hydroxycinnamoyl-CoA shikimate (HCT), cinnamoyl-CoA reductase (CCR), cinnamyl alcohol dehydrogenase (CAD; EC1.1.1.195), caffeoyl-CoA O-methyltransferase (CCOAOMT), and lactoperoxidase enzyme (LPO), involved in monolignols synthesis such as p-hydroxyphenyl (H), guaiacyl (G) and syringyl (S). Among them, the HCT, CCOAOMT, and CAD showed a higher expression in all the samples, except the IVL for the CAD, while CCR showed a higher expression in the WC. The gene expression differences between the tissue samples for cinnamic acid 4-hydroxylase (C4H), ρ-coumarate 3-hydroxylase (C3H), ferulate 5-hydroxylase (F5H), caffeic acid O-methyltransferase (COMT), and cinnamyl alcohol dehydrogenase (CAD) were below the threshold of the FPKM without any differential expression in the studied samples. 

### 2.7. DNA Methylation Analysis Using Bisulfite Sequencing

Genome wide methylation percentages determined from the bisulfite sequence data from the leaf and four tissue cultured samples, were higher in all methylated cytosine contexts for the samples from the EVL (CpG 73.2%, CHG 36.2% and CHH 33.7%) and IVL (CpG71.3%, CHG 35.4% and CHH 33.5%). The lowest levels were for the EC (CpG 53.4%, CHG 18.5% and CHH 25.3%), followed by the WC (CpG 63.8%, CHG 21.9% and CHH 28.1%) and the DC (CpG 68.4%, CHG 25.9% and CHH 28.6%). We also evaluated the DNA methylation levels of three groups of genes (30 genes in total) including DNA methyltransferase-related genes across the genome of *B. rotunda* (Figure 6a–d). In general, the CHH methylation levels were higher than the methylation levels in the CpG and CHG context, and out of 30 genes, 22 genes (73.3%) showed low methylation levels (<0.1) in the CHG and CHH cytosine contexts, whereas only 30% of the genes showed low methylation levels in the CpG context. Cytosine methylation of the methylation-related genes in all cytosine contexts (CpG, CHG and CHH) was the highest for the DRM2, followed by MET1 and CMT3 (Figure 6a–d). Among the somatic embryogenesis-related genes, the WOX gene was heavily methylated in the CpG and CHG contexts compared to the other somatic embryogenesis-related genes (LEC2, BBM, SERK) (Figure 6a,d). For the pathway-related genes, the LPOs methylated more in all the studied samples compared to the other genes.

### 2.8. Correlation between Gene Expression Levels and DNA Methylation Levels of Genes Related to Methylation, Somatic Embryogenesis and Secondary Metabolite Pathway

From the gene expression analysis, we observed that the expression level of the DNA methyltransferase genes, MET 1, CMT 3, and DRM2 was higher in the callus than the leaf samples and was highest in the embryogenic callus (EC) for all three genes, but lowest in the in vitro leaf (IVL) (Figure 7a). DRM2 showed the lowest level of expression and the highest level of DNA methylation. The DNA methylation levels of these genes at the CpG, CHG, CHH cytosine contexts were the highest in the DC and WC with similar and lower methylation levels in the embryogenic callus and leaf samples. Overall, the expression of methylation-related genes was higher in the samples of EC, DC, and WC but other than for the CMT3, which showed an inverse relationship between the expression level and methylation levels, there was no clear correlation between the level of DNA methylation and the level of gene expression (Figure 7b).

Similarly, while there were different expression patterns for the four somatic embryogenesis-related genes of SERK, BBM, LEC2, and WOX between the different leaf and callus samples (Figure 7c), the DNA methylation level of each gene across the different leaf and callus samples was largely unchanged (Figure 7d). A comparison of 23 of the *B. rotunda* genes involved in the flavonoid and phenylpropanoid pathways showed them to be expressed differentially in the *B. rotunda* leaf and callus samples. Among them, the BGLU, CAD, CHS, LPO8, LPO9 and PAL were expressed more highly in the callus than in the leaf samples (Figure 7e). The highest level of DNA methylation was observed for the HCT, CCR, and LPO2 genes in all the studied samples and again, there was no general correlation between the gene expression levels and methylation levels for these samples (Figure 7f).

## 3. Discussion

We present a genome assembly of *Boesenbergia rotunda* (2n = 36) with an estimated genome size of 2.4 Gb. The genome of the plant we sequenced, when in cultivation a largely vegetatively propagated species, shows an unusually high heterozygosity of 3.01%, suggesting that the cultivar may be of hybrid origin or may have undergone whole genome duplication events. This is also suggested based on the large number of unigenes, 408 in *B. rotunda*, notably more than twice that of *Ensete glaucum* [57], and 46,765 duplicated genes (65.8% of the *B. rotunda* genome, with at least 50% support). As noted in Citrus limon [65], high levels of heterozygosity complicate the assembly process and due to the clonal propagation nature of the fingerroot ginger, offspring resulting from sexual hybridization is rather limited. Thus, we applied a similar approach as reported by Chin et al. (2016) and Baek et al. (2018), for the assembly of the *B. rotunda* genome [66,67]. The sequencing assembly of *B. rotunda* using long PacBio reads, in addition to the Illumina short-reads, and followed by assembly using the FALCON assembler resulted in a scaffold number of 10,627. This relatively high scaffold number is not unexpected considering the high repeat content (72.51%) of the *B. rotunda* genome, coupled with the relatively high level of heterozygosity (3.01%), and the lack of any molecular marker and breeding data for *B. rotunda*. Future mapping and marker studies could help to resolve an assembly into the anticipated 18 chromosomes, as could more recent technologies such as single chromosome sequencing and optical mapping [56]. 

Sequence information for other *Boesenbergia* species is not yet available, with the closest relative of *B. rotunda* from sequenced genomes at the time of our study being M. acuminata, based on previous analyses using amino acid data from single genes including chalcone isomerase (CHI) [68] and phytyltransferase (BrPT2) [69]. Our phylogeny analysis also showed M. acuminata as the closest relative among those compared with *Z. mays*, *O. sativa*, *H. vulgare*, and *B. distachyon* from the Poaceae family, though more distantly related, as expected. 

The repeat content of the *B. rotunda* at ~72% of the assembled genome is high compared to many other plant genomes in this order such as *Musa itinerans* (38.95%) [70] and *M. acuminata* (35.43%) [71], but similar to that of *Z. officinale* (ginger official) at 81% [72]. A higher level of repeat content has been observed to correlate with the larger genome sizes in Fabaceae [73] and Melampodium [74]. Both of those reports suggest the greater genome size to be largely driven by Ty3/gypsy LTR-retrotransposons and it is interesting to note that *B. rotunda* also has a high LTR content of 64%. While data for the genome sizes and content are not yet available for other *Boesenbergia* species, the *Z. officinale* genome has a similarly high value of 61% LTR which was also suggested to contribute to the high genome size [75]. Studies in other plant species have reported that plant genomes generally have over 50% transposable elements content (e.g., maize) while some small plant genomes such as *Arabidopsis* may have as low as 10% repeat content [76,77,78]. Cytosine methylation is usually much denser in transposons than in genes [79,80,81] and this has also been correlated with the evolution of genome size in angiosperms [77]. The large genome size and high repeat content of *B. rotunda* with relatively low gene body cytosine methylation levels of the genes selected for observation in the current study, fit well with this model and it will be interesting to compare this with other *Boesenbergia* species in the future when similar data becomes available.

As DNA methylation is dynamic, we saw variations in the global DNA methylation levels in the different samples. Unmethylated DNA has been shown to demarcate expressed genes [82] and so to be able to examine this in the context of gene expression in *B. rotunda* and to add depth to our genome data, we included deep sequencing of the leaf and callus transcriptomes from *B. rotunda*. There are several alternative tools for the de novo assembly of RNA-seq short reads into a reference transcriptome and we compared analysis from four assemblers. The quality of the assembly was noticeably affected by both the k-mer size and assembler tool, with Oases delivering the highest N50 size and average contig length at k-mer 21 compared to at k-mer 24 or other assemblers (Appendix A, Table 3), indicating a more effective and accurate assembly. In comparison to a previous transcriptome assembly of *B. rotunda* by the SOAPdenovo-Trans de novo assembler, our study obtained a longer N50 size (1019) compared to an N50 value of 236 reported by [40]. An Oases assembly of the genome sequence data from a Fern, *Lygodium japonicum*, was also found to give the best mean transcript length and N50 size when compared to assemblies using Trinity and SOAPdenovo-Trans [83]. The BUSCO assessment of the *B. rotunda* transcriptome data also showed that Oases had higher numbers of complete and single copy contigs and less fragmented contigs. Based on this, the transcriptome assembly using Oases offered an improved resource for genome annotation and the gene expression study in *B. rotunda*. 

We focused the functional aspects of the *B. rotunda* genome study on the methylation and the flavonoid and phenylpropanoid pathways, as the chalcone, panduratin A, is considered one of the most promising bioactive compounds from *B. rotunda* and previous studies from our research group had indicated that DNA methylation may influence the gene expression in tissue cultured materials [84,85]. From the 23 flavonoid and phenylpropanoid pathway genes that showed differential expression between the leaf and any of the callus samples, most were more highly expressed in the EC, DC, and WC, including PAL, CHS, CHI, DFR, BGLU, HCT, CCOAOMT, and CAD (Figure 7), with the highest expression level in the non-embryogenic callus (DC and WC). This aligns with previous ultra performance liquid chromatography-mass spectrometry (UPLC-MS) data showing the WC followed by the DC to have a higher concentration of panduratin, pinocembrin, pinostrobin, cardamonin and alpinetin [43]. Based on this, the unigenes identified in the genome assembly that correspond to CHS and CHI, encode key enzymes in the biosynthesis of panduratin A in *B. rotunda*. Although DNA methylation plays an important role in the regulation of gene expression, comparison of the methylation of the differentially expressed flavonoid and phenylpropanoid pathway genes, with their cytosine methylation, showed no obvious patterns to indicate any correlation for this gene set. 

As our samples included embryogenic and non-embryogenic callus tissue, we also evaluated the expression level of DNA methylase genes (MET1, CMT3, DRM2) and genes related to somatic embryogenesis (SERK, BBM, LEC2, WUS) with the DNA methylation levels across the genome of *B. rotunda* based on a bisulfite sequence analysis. An earlier study with some quantitative qRT-PCR validation suggested that the higher level of expression of the methyltransferase-related genes and the lower CG, CHG and CHH sequence contexts in the EC samples was negatively correlated with the total methylation level of the DNA methyltransferase-related genes [85]. We did observe a similar pattern for CMT3 in all five sample types in the current study (Figure 7); however, no similar correlation between the expression level and cytosine methylation was observed in the current data for the other genes examined. The lack of correlation between the transcript expression and the respective gene body methylation from our data may be due to the limitations of the current genome assembly, such that the cis regions could not be well annotated. In the future, a higher resolution genome assembly for *B. rotunda* would be useful to examine the methylation data from the current study.

Although only a minor portion of the *B. rotunda* genome at around 0.35%, microsatellites are the key elements in plant genomes. Among these, short sequence repeat microsatellites (SSRs) have found wide utility as co-dominant markers useful in breeding and diversity studies [86,87]. In this study, we identified the genomic and EST-SSRs from *B. rotunda*, designing primers and showing several to have a transferability to *Musa* and *Ensete* genomes, mostly in silico analysis, but with 14 tested in PCR experiments. *Boesenbergia*, *Musa* and *Ensete* are members of the same plant family Zingiberales, and all have abundant AT-rich SSR sequences; however, they are not from the same genus, so are phylogenetically somewhat distant as reflected in the fairly low numbers with potential as markers across these species. Nevertheless, these newly developed SSR markers enhance the genetic resources for *B. rotunda* as well as the plant family Zingiberales, and these markers could be utilized for genotyping, population structure analysis, association studies and cultivar identification as well as any other breeding application of the *Boesenbergia* spp.

In conclusion, the genome assembly of *B. rotunda* covers some 2300 Mbp divided among 18 relatively similar submetacentric chromosomes. The cultivated accession sequenced was highly heterozygous. The genome assembly, transcriptome, gene expression, SSR analysis and DNA methylation data from this study are resources that will allow further understanding of the unique secondary metabolite properties and their biosynthetic pathways in the genus *Boesenbergia* and for functional genomics of *B. rotunda* characteristics. As this data represent a first report of a *Boesenbergia* genome, this, together with the existing data from sister families such as Musacea, can contribute to the understanding of the pan-genome of the Zinzerberales, evolution of the ginger plant family and the potential genetic selection or improvement of gingers.

## 4. Materials and Methods

### 4.1. Ethics

The conduct of this study was approved by the University of Malaya’s grant management committee, chaired by Professor Noorsaadah Abdul Rahman (noorsaadah@um.edu.my), the Director of the Institute of Research Management and Monitoring, and no human, animal, or endangered or protected plant species were used as materials.

### 4.2. Plant Materials and Samples Preparation

Rhizomes of *B. rotunda* (L.) Mansf. were collected from a commercial farm in Temerloh, Pahang, Malaysia (Latitude: 3.27° N, Longitude: 102.25° E) and propagated in the laboratory using the steps outlined by Karim et al. (2018b) [85]. To promote rhizome sprouting, first the rhizomes were cleaned under running tap water for ten minutes, then they were allowed to air dry before being planted in the black polybags.

Each day, water was sprayed on the samples to induce the growth of shoots and leaves. Four weeks after planting, young ex vitro leaf (EVL) samples were collected from the rhizome-derived plants. To generate the callus samples, meristematic block explants were subcultured on MS medium supplemented with 30 g L^−1^ of sucrose and 2 g L^−1^ of Gelrite^®^ with 2,4-dichlorophenoxy acetic acid (2,4-D) at concentrations of 1 mg L^−1^ (4.5 µM) for the watery callus (WC), 3 mg L^−1^ (13.5 µM) for the embryogenic callus (EC) and 4 mg L^−1^ (18 µM) for the dry callus (DC). The WC, EC and DC samples were collected after four weeks on the respective media (eight weeks after initial culturing from the explant). Plants grown from the embryogenic calli were cultured on regeneration media (MS0) and the in vitro leaves (IVL) were collected after 8 weeks, when the leaves were large enough for nucleic acid extraction (16 weeks after the initial culturing from meristematic block explants) [84].

### 4.3. DNA Extraction and Sequencing for Genome and Bisulfite Sequence (BS-Seq) Analysis

Total genomic DNA was isolated using a modified cetyl trimethyl ammonium bromide (CTAB) method from the ex vitro leaf (EVL) of *B. rotunda* [88]. The quality and quantity of the extracted genomic DNA were determined by a NanoDrop 2000 Spectrophotometer (Thermo Fisher Scientific Inc., Waltham, MA, USA) and Qubit^®^ 2.0 Fluorometer (Thermo Fisher Scientific Inc., Waltham, MA, USA). The DNA sample was sent to BGI Shenzhen (Shenzhen, China) for library construction and de novo sequencing on the Illumina HiSeq2000 and HiSeq2500 platforms (Illumina Inc., San Diego, CA, USA) and the PacBio RS II platform (PacBio Inc., Menlo Park, CA, USA) [89]. The BS sequence analysis was performed using the extracted DNA from the EVL, EC, DC, WC, and IVL treated by sodium bisulfite. The paired-end reads were generated using an Illumina HiSeqTM 2000 platform (Illumina Inc., San Diego, CA, USA) by Sengenics Sdn. Bhd., Malaysia from a total of five samples with three biological replicates for each sample. 

### 4.4. RNA Extraction and Sequencing for Transcriptome (RNA-Seq) Analysis

The total RNA was extracted from the EVL, EC, DC, WC, and IVL samples using a CTAB method [90] and three independent sets of RNAs for each sample were generated. High quality RNA samples were sent to BGI-Shenzhen (Shenzhen, China) for library construction and sequencing using the Illumina Genome Analyzer IIx (GAIIx) platform (Illumina Inc., San Diego, CA, USA) to generate single-end reads.

### 4.5. Determination of Chromosome Number and Location of 45S and 5S rDNA Sites on Metaphase Chromosomes of B. rotunda (2n = 36) Using Fluorescent In Situ Hybridization (FISH)

The FISH technique was performed according to Schwarzacher and Heslop-Harrison (2000) [91]. ‘Fingers’ of *B. rotunda* were placed in shallow dishes with soil to initiate root growth and kept in the glasshouse at the University of Leicester, UK. Newly grown root tips of 1–2 cm length were treated with 2 mM 8-hydroxyquinoline at growth temperature for 2 h followed by incubation overnight at 4 °C, and then fixed with 96% ethanol:glacial acetic acid (3:1). The roots were digested for 1–3 h at 37 °C with a mixture of cellulose (32 U/mL, Sigma-Aldrich, St. Louis, MO, USA, C1184), ‘Onozuka’ RS cellulose (20 U/mL), pectinase (from Aspergillus niger, Sigma-Aldrich P4716) and Viscozyme (20 U/mL, Sigma-Aldrich V2010) in a 10 mM citric acid/sodium citrate buffer (pH4.6). Chromosome preparations of dissected meristems were made in 60% acetic acid by squashing under a cover slip. Slides were stored at −20 °C until the FISH.

The 45S rDNA and 5S rDNA probes were labelled by random priming (Invitrogen Inc., Carlsbad, CA, USA) with digoxigenin 11-dUTP or biotin 11-dUTP (Roche, Basel, Switzerland) using the linearised clone pTa71 or the PCR amplified insert of clone pTa794 (from Triticum aestivum, Gerlach and Bedbrook 1979), respectively [92]. For hybridization, 50–100 ng of the labelled probes were prepared in a 40–50 µL mixture of 40% (*v*/*v*) formamide, 20% (*w*/*v*) dextran sulphate, 2xSSC (sodium chloride sodium citrate), 0.03 μg of salmon sperm DNA, 0.12% SDS (sodium dodecyl sulphate) and 0.12 mM EDTA (ethylenediamine-tetra acetic acid). The chromosomes and probe mixtures were denatured together at 70 °C for 6–8 min, before cooling down slowly to 37 °C and hybridized for 16 h at 37 °C. The slides were washed at 42 °C in 0.1xSSC and the hybridization sites were detected with anti-digoxigenin-FITC (2 µg/mL; Roche) and Streptavidin-Alexa594 (1 µg/mL; Molecular Probes Inc., Eugene, OR, USA). The chromosomes were counterstained with DAPI (4’,6-diamidino-2-phenylindole, 4 µg/mL) and mounted in CitifluorAF. The slides were examined with a Nikon Eclipse 80i microscope and images were captured using NIS-Elements v2.34 (Nikon, Tokyo, Japan), and a DS-QiMc monochrome camera. The images were pseudocoloured and the final figures were prepared with Adobe Photoshop CC2018 using enhancements that treat all pixels of the image [90].

### 4.6. Genome Size Estimation

Flow cytometry was used to determine the genome size of the *B. rotunda* as an unknown sample on a MACSQuant Analyzer (Miltenyl Biotec Inc., Bergisch Gladbach, Germany), using soybean (Glycine max cv. Polanka (G) 2C = 2.50 pg DNA) and pea (Pisum sativum cv. Ctirad (P) 2C = 9.09 pg DNA) as the internal standards and propidium iodide as the stain. Each plant (sample and comparator) was compared using an average of four biological replicates [53,93,94]. We also performed a K-mer analysis to estimate the *B. rotunda* genome size and heterozygosity rate using Jellyfish [95] and GenomeScope [96]. 

### 4.7. Genome Assembly

A combination of sequencing technologies of the PacBio RSII platform, Illumina HiSeq 2500 paired-end reads (PE) with a 450 bp insert size library, and Illumina HiSeq 2000 mate-pair reads (MP) with insert size libraries of 2, 5, 10, 20, and 40 kb was performed for the genome assembly. Before assembly, the Illumina HiSeq sequence reads were filtered by removing the adaptors and low-quality nucleotides. PacBio reads were filtered to remove the short reads of less than 500 bp or a quality score lower than 0.8, then error correction for the long reads was completed by FALCON [67,97], following the general principles proposed by [98]. We tried to use several de novo assemblers to construct the assembly with both the Illumina and PacBio reads. Finally, we chose the SMARTdenovo software [99]. Corrected PacBio reads were assembled with the SMARTdenovo software (https://github.com/ruanjue/smartdenovo accessed on 1 June 2018) to construct the contigs. For the PacBio data, constructed contigs were subsequently polished by stand-alone consensus modules [98] and Pilon software [100] for the Illumina PE reads. Polished contigs were used as input for the scaffolding. The scaffolds were constructed by SOAP scaffolding, and the SSPACE tool [101] with Illumina mate-pair reads (2–40 k) with default parameters to extend the length of the scaffolds for the raw assembly. The gaps within the scaffolds, and consensus sequences generated from the PacBio sub-reads were filled using PBJelly2 [102]. Finally, the scaffolds were corrected by Pilon [100] with Illumina PE reads to correct the assembly errors and to obtain the final genome assembly.

The completeness of the genome assembly was tested by searching for 1440 core eukaryotic genes using benchmarking universal single-copy orthologs (BUSCO) (v2.0) [61]. The quality of the genome assembly was assessed by mapping of the Illumina paired-end 250 bp read data to the contigs using BWA-MEM (version 0.7.15-r1142) [103]. 

### 4.8. Repeat Annotation

Tandem repeats were identified with the tandem repeat finder (TRF) [104] (version 4.0.4). The transposable elements (TE) were identified with integrated homology-based and de novo methods [53]. The homology-based prediction was completed at the DNA and protein levels by comparing the assembly to the RepBase v.20.04 [105] database as a query library using RepeatMasker v.4.0.7 (http://www.repeatmasker.org/ accessed on 1 June 2018) and ProteinRepeatMask v.4.0.7 (http://www.repeatmasker.org/ accessed on 1 June 2018). To search those absent TEs in the RepBase library, a de novo repeat library was constructed using RepeatModeler v.1.0.10 (http://www.repeatmasker.org/ accessed on 1 June 2018) to run against the *B. rotunda* genome assembly using RepeatMasker v.4.0.7 (http://www.repeatmasker.org/ accessed on 1 June 2018). 

### 4.9. Gene Annotation 

The protein-coding genes prediction was completed by homology, de novo, and RNA-Seq-based approaches. For generation of the homology-based predictions, the gene sets from four species, i.e., *M. acuminata* (http://www.promusa.org/Musa+acuminata accessed on 28 May 2019), *P. dactylifera*, *O. sativa* (http://rice.plantbiology.msu.edu/ accessed on 28 May 2019) and *A. thaliana* (https://www.arabidopsis.org/ accessed May 2019) were downloaded. The nonredundant protein sequences for each gene set was searched by TBLASTN. For generation of the expression-based evidence, RNA-Seq short reads originated from the EVL, IVL, EC, DC, and WC tissues were mapped to the ginger genome with the Hisat2 v.2.0.4 [106] alignment program. For the de novo gene annotation, well-supported transcripts identified both by the homology-based and the RNA-Seq based predictions were selected for ab initio prediction using AUGUSTUS v.3.2.3 program [107,108]. The exon–intron structure of the genes was predicted using Genscan [109] and SNAP [110]. The results from the three approaches were consolidated using MAKER v.2.31.9 [111] to generate a protein-coding gene set. For the functional information, in silico translated products of the coding genes were aligned to the seven known protein databases of NR [112], InterPro [113], GO [114], KEGG [115], Swissprot and TrEMBL [116], COG [117].

### 4.10. Non-Coding RNAs Annotation

Four types of ncRNA, including microRNA (miRNA), transfer RNA (tRNA), ribosomal RNA (rRNA), and small nuclear RNA (snRNA), were annotated in the assembled genome based on de novo/or homology-based search methods. The tRNAs genes were annotated using tRNAScan-SE v.1.3.1 [118] with default parameters and filtered to remove the pseudo annotated tRNA genes. To identify the rRNA genes, the *B. rotunda* genome assembly was searched against the rRNA template sequences (Rfam database, release 13.0) [119] of A. thaliana, ca and M. acuminata with BLASTN, with an identity cutoff of ≥90% and a coverage at 80% or more. Using Infernal v.1.1.2 [120], mapping of the *B. rotunda* genome sequences to the Rfam database was performed to identify the miRNA and snRNA genes [53,89].

### 4.11. Construction of Phylogenetic Trees

The conserved orthologs genes (COS) in the *B. rotunda* genome and 13 other species were identified using the Orthofinder program [62]. Using identified single-copy orthologous genes, a neighbour joining (NJ) tree was constructed using a MEGAX [121] and UpSet plot using UpSetR [122].

### 4.12. Gene Family Expansion and Contraction Analysis

The data generated from OrthoFinder were used as input for the computational Analysis of gene Family Evolution (CAFE) [123] to estimate the gene family expansion and contraction. The phylogenetic tree from OrthoFinder was converted to an ultrametric tree using the make_ultrametric.py in OrthoFinder. Gene families with a large variance (≥100 gene copies) were removed using the clade_and_size_filter.py in the CAFE package. Divergence times in the phylogenetic tree were estimated using PATHd8 [124] and were calibrated using the divergence time between *Brachypodium* and *Oryza* (40–45 million years ago) (The International Brachypodium Initiative 2010) [125] and *Arabidopsis* and *Oryza* (130–200 million years ago) [126]. CAFE version 5 [123] was used to determine the stochastic birth and death processes and for modelling of the gene family evolution. The parameters for CAFE5 were “cafe5 -i orthofinder_gene_families.txt -t orthofinder_ultrametric.tre -p -e”.

### 4.13. DNA Methylation Analysis Using Bisulfite Sequencing (BS-Seq)

For the bisulfite sequencing analysis, low quality reads and adapters were trimmed by the Trim-Galore [127] tool specific for bisulfite sequencing. The trimmed reads were mapped to draft the ginger genome with the Bismark [128] tool by choosing the Bowtie aligner with options set to the best, minimum map length of 50 bp and insert size of 500 bp. Mapping duplicates were removed by the Methpipe [129] tool. The Methcounts program from the Methpipe software package was used for the mapping of methylated and unmethylated cytosines where the methylation level at a single base resolution was calculated based on the number of 5-methylated cytosines (5 mC) in the reads, divided by the sum of the C and thymines (T) in the CG, CHG and CHH sequence contexts within the coding sequences of all the selected genes from *B. rotunda* [84].

### 4.14. De Novo Transcriptome Assembly of B. rotunda and Functional Annotation

To gather information related to the secondary metabolites, expression of the genes involved in the flavonoid and phenylpropanoid pathways of *B. rotunda* was based on the deep transcriptome sequencing of three cell culture types, in vitro and ex vitro leaves of *B. rotunda*. Based on our previous studies of embryogenesis related genes [84,85] and on the levels of metabolites in the cell cultures [43], we generated deep transcriptome data from five tissue types (each with three replicates) to investigate the gene regulation patterns in the phenylpropanoid and flavonoid pathways, to identify the metabolite producing cells in the *B. rotunda* in vitro cultured cells. The quality of the RNA-Seq raw reads were checked by the FastQC software (https://www.bioinformatics.babraham.ac.uk/projects/fastqc/ accessed on 28 May 2018) to remove the low-quality reads and adapter sequences. Four different short-read assemblers: Oases [130], TransABySS [131], SOAPdenovo-Trans [132] and Trinity [133], were used to assemble the high-quality reads to the contigs. Using a best k-mer strategy, the high-quality reads were assembled at different k-mer lengths 21–51 using Oases, TransABySS and SOAPdenovo-Trans, whereas the assembly by Trinity used default parameters (K-mer 25). The contigs generated from all the k-mers by each respective assembler were merged and the redundancy was removed using CD-HIT [134]. Then, the TGICL clustering tool [135] was used to assemble the non-redundant contigs with a maximum identity of 90 and a minimum overlap length of 40. The completeness of the transcriptome assemblies was measured using the BUSCO [61] software.

High-throughput functional annotation was performed with the Blast2GO Command Line [136]. The BLASTX algorithm was performed to obtain a list of the potential homologous for each input sequence. To evaluate the functional annotation for the query sequences [137], gene ontology (GO) terms associated with the obtained BLAST hits were mapped by Blast2GO. GO mapping and Enzyme Commission (EC) classification were performed based on the annotation parameters of a Cut-off 55, E-Value 1 × 10^e−6^, GO Weight of 5, and HSP-Hit Coverage Cut-off 0. In order to identify the functional enrichment categories among the differentially expressed genes (DEGs), a Fisher exact test with a false discovery rate (FDR) cutoff of 0.05 was used. Classification of the *B. rotunda* transcripts into functional categories was performed using the Eukaryotic Orthologous Groups (KOG) [117] protein database. Biological pathways mapping of the *B. rotunda* transcripts was completed using the KEGG database [115]. Unigenes potentially related to the Panduratin A and other secondary metabolites biosynthesis were identified as those with a unigene annotated function matching to the enzymes assigned to the flavonoid and phenylpropanoid biosynthesis pathways in the KEGG pathway database.

### 4.15. Estimation of Transcript Abundance and Differential Expression

The RSEM software package [138] was used for the estimation of the gene expression level with a mean fragment length of 200 bp and fragment length standard deviation of 80 bp. Then, the FPKMs (fragments per feature kilobase per million reads mapped) were used to normalize the expression level for each gene and comparison between the samples. The Bioconductor tool (EdgeR) [139] was used for the differential expression analysis with a *p*-value threshold of ≤0.05 and |log2 (Fold Change)| ≥1 used to identify the significant differential expression of the transcripts.

### 4.16. Mining of Simple Sequence Repeats (SSRs) from B. rotunda Transcriptome and Genome Assembly

The whole genome assembly and assembled transcriptome sequences were searched for SSRs using a modified Liliaceae simple sequence analysis tool (LSAT) pipeline [140]. The searches were standardized for mining SSRs from mono to 20 bp with a minimum repeat loci of 12 nucleotides. The SSRs were classified based on the SSR locus length (class I > 20 nt and class II 12–20 nt) and the nucleotide base composition of the SSR loci (AT-rich, GC-rich and AT-GC balance). Primer pair sequences were developed for each identified SSR loci using the default parameters of the primer 3 (http://bioinfo.ut.ee/primer3 accessed on 28 May 2020) software [141]. Redundant primer pairs were eliminated using the perl script developed by Biswas et al. [87]. An electronic polymerase chain reaction (ePCR) [142] strategy was applied for mapping and estimating the transferability of the designed primers. The primers were mapped on four genomes viz. *M. acuminata*, *M. balbisiana*, *M. itinerans* and *E. ventricosum*, as those are the most related plant species of *B. rotunda*. A maximum 2 nt mismatch with two gaps was set as a cut off value for the ePCR result filter.


*Wet lab validation of the transcriptome SSR (EST-SSR) and genomic SSR (G-SSR) markers*


A total of 14 (8 EST-SSR and 6 G-SSR) primer pairs were selected based on their in silico transferability result to assess their marker potentiality. Three *B. rotunda*, two Ensete and three Musa species were used to validate the selected primer sets. Fresh leaf samples were harvested from the greenhouse grown plants and the total genomic DNA was extracted following the CTAB methods. PCR amplifications were carried out for the SSR primer validation under the following conditions: 95 °C for 2 min, 35 cycles at 95 °C for 1 min, 60 °C for 1 min, and 72 °C for 1 min, followed by a final elongation at 72 °C for 10 min. The amplified DNA fragments were run on 2% agarose gels in a 1 × Tris–Borate-EDTA (TBE) buffer with 80v for 90 min. A 100 bp molecular ladder was used to estimate the amplicon size.

## Figures and Tables

**Figure 1 ijms-23-07269-f001:**
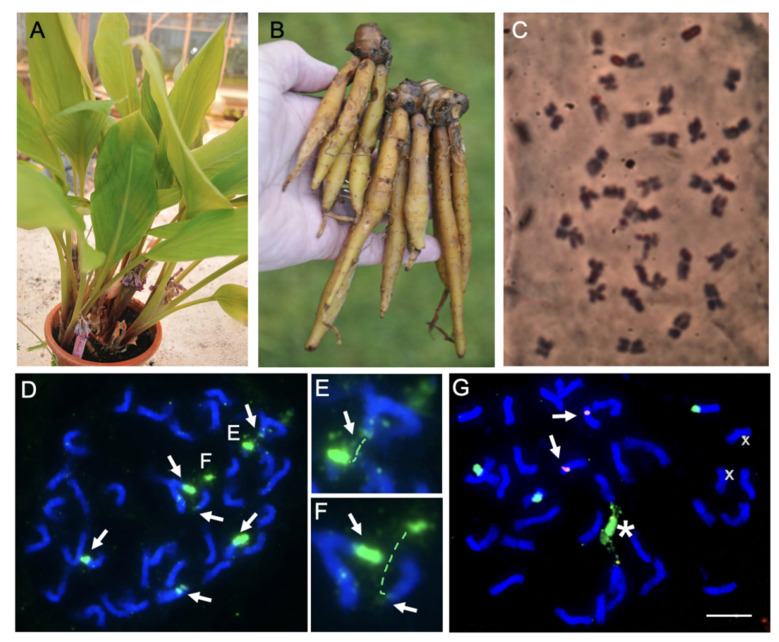
Boesenbergia rotunda leaves, rhizomes and number and location of 45S and 5S rDNA sites on metaphase chromosomes (2n = 36). (**A**) Leaves. (**B**) Rhizomes. (**C**) Chromosome preparation using fresh root tips. (**D**–**G**) Fluorescent in situ hybridization with clone pTa71 (45S rDNA of wheat), labelled with digoxigenin and detected with FITC (green) and clone pTa794 (5S rDNA of wheat) labelled with biotin and detected with Alexa 674 (shown in red). (**D**–**F**) early metaphase showing six sites of 45S rDNA (arrows) of variable strength at ends of 3 pairs of chromosomes. In some cases, the rDNA is extended, and the satellite is separated from the main chromosomes shown enlarged in (**E**,**F**). (**G**) Two 5S rDNA sites (arrows) were detected on a chromosome pair not bearing 45S rDNA. The star indicates the fusion of 2 or 3 45S rDNA sites.

**Figure 2 ijms-23-07269-f002:**
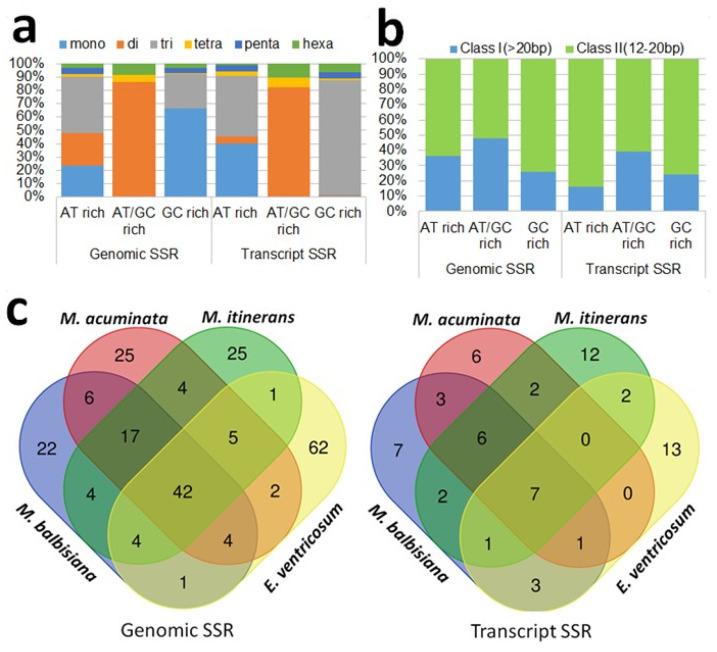
(**a**,**b**) frequency distribution of SSR motif; (**c**) transferability of genomic and transcript SSR markers in four relatives of *B. rotunda*.

**Figure 3 ijms-23-07269-f003:**
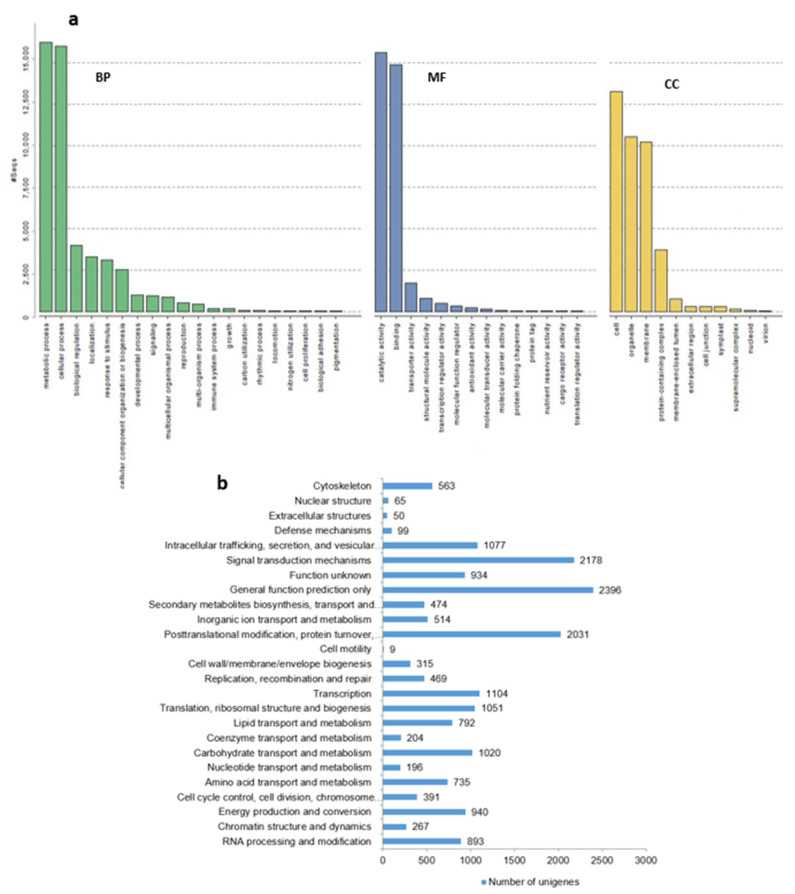
(**a**) Gene ontology (GO) classification of assembled unigenes of *B. rotunda*. Results are summarized in three main categories: biological process (BP), molecular function (MF), and cellular component (CC). The *x*-axis indicates the subgroups in the GO annotation while the *y*-axis indicates the percentage of specific categories of genes in each main category; (**b**) distribution of Eukaryotic Orthologous Groups (KOG) classification. A total of 18,767 assembled unigenes were annotated and assigned to 25 functional categories. The vertical axis indicates subgroups in the KOG classification and the *x*-axis represents the number of genes in each main category.

**Figure 4 ijms-23-07269-f004:**
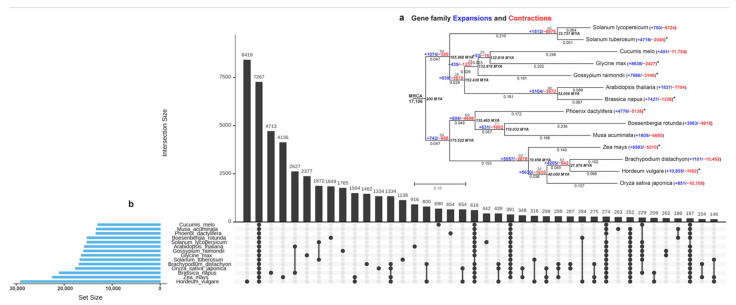
(**a**) cross-genera phylogenetic analysis of *B. rotunda* and 13 other species. Aterisks after the expansion and contraction figures indicate genomes with reported ancient whole genome duplication or massive segmental duplications or major chromosomal duplications; (**b**) UpSet plot showing unique and shared protein ortholog clusters of *B. rotunda* and 13 selected reference genomes. Connected dots represent the intersections of overlapping orthologs with the vertical black bars above showing the number of orthogroups in each intersection.

**Figure 5 ijms-23-07269-f005:**
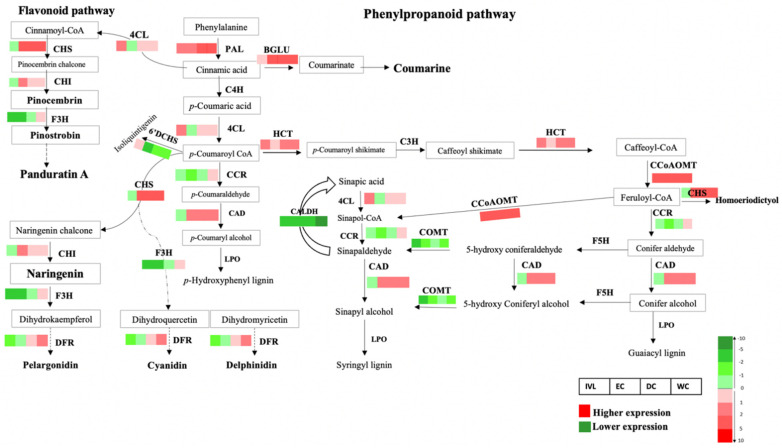
Scheme of the flavonoid and phenylpropanoid biosynthetic pathways in *B. rotunda* based on KEGG pathways. Genes encoding enzymes for each step are indicated as follows: CAD, cinnamyl alcohol dehydrogenase; BGLU, Beta-glucosidase; CALDH, coniferyl-aldehyde dehydrogenase; C4H, cinnamic acid 4-hydroxylase; 4CL, 4-coumarate–CoA ligase; CHS, chalcone synthase; CHI, chalcone isomerase; CCoAOMT, caffeoyl-CoA 3-*O*-methyltransferase; C3H, ρ-coumarate 3-hydroxylase; CCR, cinnamoyl-CoA reductase; COMT, caffeic acid *O*-methyltransferase; 6’DCHS, 6′-deoxychalcone synthase; DFR, dihydroflavonol 4-reductase; F3H, flavonoid 3-hydroxylase; F5H, ferulate 5-hydroxylase; HCT, Hydroxycinnamoyl-CoA shikimate; LPO, Lactoperoxidase; PAL, phenylalanine ammonia lyase. Beside each enzyme, four boxes are shown (from left to right): in vitro leaf (IVL), embryogenic callus (EC), dry callus (DC) and watery callus (WC). Red boxes indicate relatively higher mRNA expression compared to the leaf sample with the highest levels in darker red. Green boxes indicate relatively lower expression compared to the leaf sample. The colour box is based on log2FC values.

**Figure 6 ijms-23-07269-f006:**
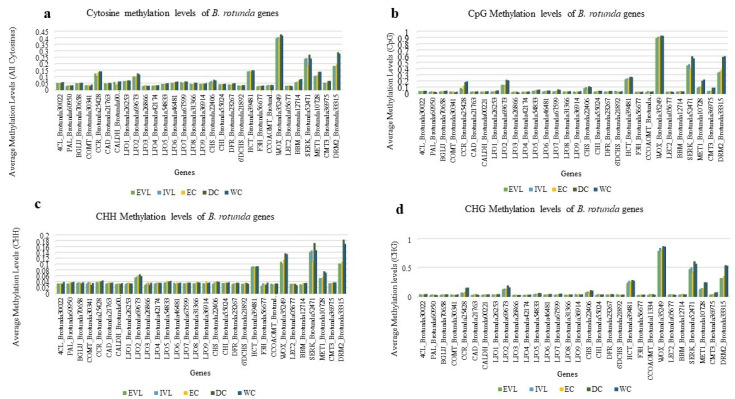
(**a**–**d**) average methylation levels of DNA methyltransferase-related genes (MET1, CMT3, and DRM2), somatic embryogenesis genes (SERK, BBM, LEC2, and WUS), and genes involved in flavonoid and phenylpropanoid biosynthesis pathways in different samples of *B. rotunda*; ex-vitro leaf (EVL), in vitro leaf (IVL), embryogenic callus (EC), dry callus (DC), and watery callus (WC). (**a**) cytosine methylation; (**b**) CpG methylation; (**c**) CHG methylation; (**d**) CHH methylation. CAD, cinnamyl alcohol dehydrogenase; BGLU, Beta-glucosidase; CALDH, coniferyl-aldehyde dehydrogenase; C4H, cinnamic acid 4-hydroxylase; 4CL, 4-coumarate–CoA ligase; CHS, chalcone synthase; CHI, chalcone isomerase; CCoAOMT, caffeoyl-CoA 3-O-methyltransferase; C3H, ρ-coumarate 3-hydroxylase; CCR, cinnamoyl-CoA reductase; COMT, caffeic acid O-methyltransferase; 6’DCHS, 6′-deoxychalcone synthase; DFR, dihydroflavonol 4-reductase; F3H, flavonoid 3-hydroxylase; F5H, ferulate 5-hydroxylase; HCT, Hydroxycinnamoyl-CoA shikimate; LPO, Lactoperoxidase; PAL, phenylalanine ammonia lyase; WOX, Wuschel; LEC3, Leafy cotyledon 2; BBM, Baby boom; SERK, Somatic embryogenesis receptor-like kinase; MET1, Methyltransferase 1; CMT3, Chromomethylase 3; DRM2, Domain rearranged methyltransferase 2.

**Figure 7 ijms-23-07269-f007:**
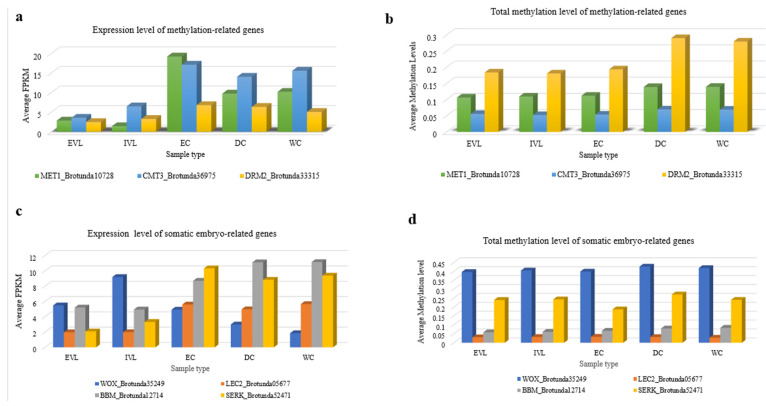
Expression level and total methylation level in all cytosine contexts of methylation-related genes (**a**,**b**), somatic embryogenesis-related gene (**c**,**d**) and flavonoid and phenylpropanoid biosynthesis pathways-related genes (**e**,**f**) in ex vitro leaf (EVL), in vitro leaf (IVL), embryogenic callus (EC), dry callus (DC), and watery callus (WC). CAD, cinnamyl alcohol dehydrogenase; BGLU, Beta-glucosidase; CALDH, coniferyl-aldehyde dehydrogenase; C4H, cinnamic acid 4-hydroxylase; 4CL, 4-coumarate–CoA ligase; CHS, chalcone synthase; CHI, chalcone isomerase; CCoAOMT, caffeoyl-CoA 3-*O*-methyltransferase; C3H, ρ-coumarate 3-hydroxylase; CCR, cinnamoyl-CoA reductase; COMT, caffeic acid *O*-methyltransferase; 6’DCHS, 6′-deoxychalcone synthase; DFR, dihydroflavonol 4-reductase; F3H, flavonoid 3-hydroxylase; F5H, ferulate 5-hydroxylase; HCT, Hydroxycinnamoyl-CoA shikimate; LPO, Lactoperoxidase; PAL, phenylalanine ammonia lyase.

**Table 1 ijms-23-07269-t001:** Statistics of the final genome assembly of the *B. rotunda*.

		Scaffolds		Contigs		
	No.	Size (bp)		No.	Size (bp)	Gaps
		With gaps	Without Gaps			
**Total Number**	10,627			27,491		16,864
**Min**	-	5830	5830	-	5198	25
**Median**	-	136,187	131,005	-	55,047	2415
**Mean**	-	220,901	213,350	-	82,473	4758
**Max**	-	2,848,924	2,758,809	-	1,033,476	38,914
**Total size**	-	2,347,517,452	2,267,274,222	-	2,267,274,222	80,243,230
**N50**	-	394,682	379,106	-	123,867	11,038
**N90**	-	107,821	103,307	-	37,045	2551
**N95**	-	69,101	66,089	-	27,170	1540
**GC content (%)**	40.1					

**Table 2 ijms-23-07269-t002:** Evaluation of completeness of the final assembly.

Species	Read Length (bp)	Data	Sequence Depth (X)	Mapped (%)	Properly Paired (%)	Singletons (%)	Reference Total Length (Gb)	Reads Covered Length (Gb)	Coverage (%)
*B. rotunda*	250_250	260 (Gb)	104	95.24	84.47	0.20	2.35	2.25	96

**Table 3 ijms-23-07269-t003:** Comparison of *denovo* transcriptome assembly results for four different assembly software: SOAP-*denovo*, Oases, TransAbyss, and Trinity.

Features	SOAP-*denovo* (K25)	Oases (K21)	TransAbyss (K25)	Trinity (K25)
N50 size (bp)	410	1019	495	487
N50 No.	22,910	14,286	28,234	36,730
Contig number	78,492	72,085	111,327	158,465
Transcript’s size (bp)	30,869,274	51,258,323	50,358,442	70,949,809
Avg. transcript length (bp)	393	711	452	448
Min. contig length(bp)	200	200	200	200
Max. contig length (bp)	15,760	12,523	33,886	13,325
Assessment assembly after merged assembly of non-redundant contigs from different k-mers via TGICL				
N50 size (bp)	572	1013	607	536
N50 no.	18,528	17,329	28,419	26,034
Contig number	78,963	95,847	132,572	115,096
Transcriptome size (bp)	38,503,434	66,535,881	67,286,353	54,131,258
Average length (bp)	488	694	508	470
Min. contig length(bp)	200	200	200	200
Max. contig length (bp)	43,900	88,053	100,968	19,761

**Table 4 ijms-23-07269-t004:** TEs Content in the assembled *B. rotunda* genome.

Type		Repeat Size (bp)	% of Genome
TRF		**162,927,183**	**6.94**
RepeatMasker (RepBase TEs)	DNA	23,107,771	0.98
	LINE	7,269,664	0.31
	LTR	308,993,979	13.16
	SINE	50,226	0.00
	Other	1305	0.00
	Unknown	0.00	0.00
	**Total**	**339,001,341**	**14.44**
RepeatProteinMask (TE proteins)	DNA	30,071,545	1.28
	LINE	15,711,684	0.67
	LTR	449,069,450	19.13
	SINE	0.00	0
	Other	0.00	0
	Unknown	0.00	0
	**Total**	**494,297,946**	**21.05**
De novo	DNA	49,253,612	2.10
	LINE	9,765,795	0.42
	LTR	1,524,782,230	64.95
	SINE	789,305	0.03
	Other	0	0.00
	Satellite	8,247,215	0.351316
	Simple_repeat	6,742,687	0.287226
	Unknown	2,202,787	0.09
	**Total**	**1,591,591,610**	**67.80**
Combined TEs	DNA	77,273,965	3.29
	LINE	23,221,220	0.99
	LTR	1,576,612,191	67.16
	SINE	832,585	0.04
	Other	1305	0.00
	Unknown	2,202,787	0.09
	**Total**	**1,653,717,174**	**70.45**
Total		1,702,210,889	72.51

Note: Repbase TEs: the result of RepeatMasker based on Repbase; TE proteins: the result of RepeatProteinMask based on Repbase; De novo: result of RepeatMasker by using library predicted through De novo; Combined: combine the results of Repbase TEs, TE proteins and De novo.

**Table 5 ijms-23-07269-t005:** Genome and transcriptome-wide microsatellite identification and characterization in *B. rotunda*.

Item	Genome-Wide	%	Transcriptome-Wide	%
Total number of sequences examined	10,627		95,847	
Total size of examined sequences (bp)	2,347,517,452		66,535,881	
Total number of identified microsatellites	238,441		4579	
Number of microsatellites containing sequences	10,381		4032	
Sequences contain more than one microsatellite	9803		384	
Microsatellites in compound formation	4309		27	
Microsatellite’s density (per Mbp)	102		69	
Class I microsatellites	82,414	35.20	949	20.85
Class II microsatellites	151,718	64.80	3603	79.15
AT rich microsatellites	176,052	75.19	2778	61.03
GC rich microsatellites	43,155	18.43	1275	28.01
AT/GC balance microsatellites	14,925	6.37	499	10.96
Mono-nucleotide repeats	68,961	28.92	1137	24.83
Di-nucleotide repeats	61,439	25.77	574	12.54
Tri-nucleotide repeats	84,932	35.62	2366	51.67
Tera-nucleotide repeats	5330	2.24	148	3.23
Penta-nucleotide repeats	9917	4.16	185	4.04
Hexa-nucleotide repeats	7862	3.30	169	3.69
Primer modelling was successful	223,678	93.81	3348	73.12
Primer modelling failed	14,763	6.60	1231	36.77
Non redundant primer	132,792	59.37	1888	56.39
No. of Primer Mapped on *M. acuminata* genome	100	0.075	30	1.59
No. of Primer Mapped on *M. balbisiana* genome	105	0.079	25	1.32
No. of Primer Mapped on *M. Itinerans* genome	102	0.077	32	1.69
No. of Primer Mapped on *Ensete ventricosum* genome	121	0.091	27	1.43
No. of primer tested	6	100	8	100
No. of primer amplified	6	100	8	100

**Table 6 ijms-23-07269-t006:** Statistics of function annotation.

Database	Number	Percent (%)
NR	71,072	97.22
InterPro	69,525	95.11
GO	45,256	61.91
KEGG	59,649	81.60
Swissprot	57,622	78.82
COG	24,851	33.99
TrEMBL	70,990	97.11
Total annotated	73,102	97.81
Unannotated	1602	2.19

**Table 7 ijms-23-07269-t007:** Non-coding RNA genes in the genome of *B. rotunda*.

Type	Copy	Average Length (bp)	Total Length (bp)	% of Genome
miRNA	213	119	25,384	0.001081
tRNA	2727	75	205,538	0.008756
rRNA	486	232	112,876	0.004808
18S	105	666	69,922	0.002979
28S	147	119	17,441	0.000743
5.8S	40	148	5931	0.000253
5S	194	101	19,582	0.000834
snRNA	2136	154	329,909	0.014054
CD-box	600	105	62,771	0.002674
HACA-box	53	134	7091	0.000302
splicing	1483	175	260,047	0.011078

## Data Availability

Raw sequence data used for genome assembly, mRNA sequencing (RNA-Seq) and whole-genome bisulfite sequencing (BS-Seq) are available at NCBI under BioProject ID PRJNA712941.

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
