# Peer review of "Genome Assembly and Analysis of the Flavonoid and Phenylpropanoid Biosynthetic Pathways in Fingerroot Ginger (Boesenbergia rotunda)"

_ijms, 2022, doi:10.3390/ijms23137269_

Round 1

Reviewer 1 Report

The authors complete the sequencing, assembly and annotation of the genome of Fingerroot ginger (Boesenbergia rotunda) using PacBio and Illumina HiSeq platforms. Combined with Fluorescence In Situ Hybridization, transcriptome, gene expression profiles and bisulfite seq, they identified the rDNA sites on chromosomes, gene expression and DNA methylation related to somatic embryogenesis and flavonoid biosynthesis. Overall, the manuscript provides an important and useful genome source for genetic and genomic research of B. rotunda and its metabolic pathways of natural products. Although the contig N50 of 123.86 kb is not long enough for a high quality of assembled genome, I understand the high level of heterozygosity (3%) is challenging for genome assembly process. In addition, this manuscript is well written. I believe this genome is of invaluable for studying the functional genomics of B. rotunda. I only have a few questions.

Major issue.

The authors focused on flavonoid and phenylpropanoid biosynthetic pathways as the biological story of the genome, because this species is a high-value culinary and medicinal plant. The culinary and medicinal properties are from the rhizome. However, why did authors studied transcriptome changes of genes related to flavonoid biosynthesis in vitro leaf (IVL), embryogenic callus (EC), and non-embryogenic calli (dry callus (DC) and watery callus (WC)), instead of rhizome?The economically important part of Fingerroot ginger is rhizome, therefore, I wonder why did not study the rhizome. The expression pattern of flavonoid biosynthetic genes as well as flavonoid accumulation among different tissues and organs, especially rhizome, are important to understand the genetic basis of economic trait in ginger.

Minors.

Since this genome is about Fingerroot ginger, the authors should provide the photo of plant and rhizomes of Fingerroot ginger in the Figures.

Line55. Boesenbergia should be italic. All genus and species scientific names throughout the manuscript should be italic.

Line 61-67. The most important bioactivity of plant flavonoid compounds is antioxidant activity. Please add the antioxidant activity of flavonoid compounds in this paragraph (Shen et al, Food Chemistry, 2022, 383:132531.)

Line 409. “notably more than twice that of Ensete glaucum 56, and 46,765 duplication events?“ What dose “46,765 duplication events” mean?

Line 543-544.  “in vitro leaves (IVL) were collected after 8 weeks.”  Why choose this stage for sampling?

Reviewer 2 Report

The manuscript (ijms-1777888) submitted by Sima Taheri et al. demonstrates the first genome assembly of Fingerroot ginger. Furthermore, they did relatively extensive analysis of the flavonoid and phenylpropanoid biosynthetic pathways in this species. The novelty of the research is sufficient though the quality of the genome assembly is not very high. Overall, the study provides enough information for researchers interested in investigating this species. The manuscript is well written, though I still spotted a few places to be changed:

1.       Throughout the manuscript, please italicize all the species scientific names.

2.       Line 105, “40” should be “[40]”;

3.       Table 3, in the transcriptome size, change 66.535,881 to 66,535,881;

4.       Line 506, change “distanced” to “distant”;

5.       The reference is a little lengthy.

Author Response

We thank the reviewer for their time and the helpful comments.

We provide our point by point response as follows:

Point 1: Throughout the manuscript, please italicize all the species scientific names.

Response 1: We have edited and corrected to ensure that all species names are italicised.

Point 2: Line 105, “40” should be “[40]”;

Response 2: We have corrected the format for the citation.

Point 3: Table 3, in the transcriptome size, change 66.535,881 to 66,535,881;

Response 3: We have corrected the format of the entry in Table 3.

Point 4: Line 506, change “distanced” to “distant”;

Response 4: We have corrected the text in line 506.

Point 5: The reference is a little lengthy.

Response 5: We do agree, however as we used a large number of methods and because this work builds on a large number of previous reports, we aimed to credit all the relevant studies in our citations. We have as far as possible used single sources as supporting citations.